# Machine Learning Model-Based Retrieval of Temperature and Relative Humidity Profiles Measured by Microwave Radiometer

Yuyan Luo [1], Hao Wu [2], Taofeng Gu [3], Zhenglin Wang [1,4], Haiyan Yue [5], Guangsheng Wu [3], Langfeng Zhu [2], Dongyang Pu [2], Pei Tang [6] and Mengjiao Jiang [1,*]

1 Plateau Atmospheres and Environment Key Laboratory of Sichuan Province, School of Atmospheric Sciences, Chengdu University of Information Technology, Chengdu 610225, China; 3210101022@stu.cuit.edu.cn (Y.L.)
2 Key Laboratory of China Meteorological Administration Atmospheric Sounding, School of Electrical Engineering, Chengdu University of Information Technology, Chengdu 610225, China; wuhao@cuit.edu.cn (H.W.); 3210307001@stu.cuit.edu.cn (D.P.)
3 Guangzhou Meteorological Observatory, Guangzhou 511430, China
4 Hainan International Commercial Aerospace Launch Co., Ltd., Wenchang 571300, China
5 Guangzhou Emergency Warning Information Release Center, Guangzhou 511430, China
6 Zhongshan Meteorological Service, Zhongshan 528400, China
* Correspondence: jiangmj@cuit.edu.cn

**Abstract:** The accuracy of temperature and relative humidity (RH) profiles retrieved by the ground-based microwave radiometer (MWR) is crucial for meteorological research. In this study, the four-year measurements of brightness temperature measured by the microwave radiometer from Huangpu meteorological station in Guangzhou, China, and the radiosonde data from the Qingyuan meteorological station (70 km northwest of Huangpu station) during the years from 2018 to 2021 are compared with the sonde data. To make a detailed comparison on the performance of machine learning models in retrieving the temperature and RH profiles, four machine learning algorithms, namely Deep Learning (DL), Gradient Boosting Machine (GBM), Extreme Gradient Boosting (XGBoost) and Random Forest (RF), are employed and verified. The results show that the DL model performs the best in temperature retrieval (with the root-mean-square error and the correlation coefficient of 2.36 and 0.98, respectively), while the RH of the four machine learning methods shows different excellence at different altitude levels. The integrated machine learning (ML) RH method is proposed here, in which a certain method with the minimum RMSE is selected from the four methods of DL, GBM, XGBoost and RF for a certain altitude level. Two cases on 29 January 2021 and on 10 February 2021 are used for illustration. The case on 29 January 2021 illustrates that the DL model is suitable for temperature retrieval and the ML model is suitable for RH retrieval in Guangzhou. The case on 10 February 2021 shows that the ML RH method reaches over 85% before precipitation, implying the application of the ML RH method in pre-precipitation warnings.

**Keywords:** microwave radiometer; radiosonde; temperature and humidity profiles; machine learning

## 1. Introduction

Atmospheric temperature and relative humidity (RH) are important parameters of the atmosphere and environment. Temperature and RH profiles with refined vertical resolution play an important role in urban meteorological forecasting [1–5]. The coastal city of Guangzhou is frequently hit by medium- and small-scale short-term weather events (such as torrential rains, typhoons and thunderstorms), which are extremely destructive and catastrophic despite the short activity time [6]. Accurate observation of the atmospheric vertical profile is fundamental for meteorological studies.

Although the traditional radiosonde data have high representativeness and reliability, the traditional observations are expensive and lack spatiotemporal resolution [7]. Ground-based microwave radiometers (MWRs) with passive remote sensing technology

can overcome these shortcomings [8]. Since an MWR has the advantages of reliable calibration method, high resolution, unmanned continuous observation and simple operation, it is becoming an important instrument for remote sensing of atmospheric vertical profiles [9]. The MWR can continuously observe temperature, relative humidity and liquid water content within 0–10 km. In recent years, these data, combined with wind profile data, have gradually become an important reference for short-impending weather forecasting [9]. It is of high scientific importance and potential value to study the inversion of atmospheric temperature and humidity profile using microwave radiometer data [10]. However, the brightness temperature (BT) data from different channels of the MWR are disturbed by precipitation and cloud factors, resulting in abnormal values [11–13]. Meanwhile, when the sun is in the observation direction of the MWR, the BT data will be abnormally increased due to the influence of solar radiation, especially those used in low latitudes [14]. Therefore, it is essential to control the quality of the MWR observation data for a better forecast [15–17].

With the development of the ground-based MWR network, it has been widely applied to the detection of atmospheric vertical profiles in the boundary layer. Improving the reliability and accuracy of MWR observations is the priority for a refined atmospheric vertical profile. Meteorologists have proposed various methods to improve the accuracy of retrieval data, such as the linear statistical method [18], the best estimate method [19], neural networks [20,21] and machine learning [22,23]. Among these methods, the neural network performs well in solving the nonlinear relationship in the model. For example, Bao et al. [24,25] used the back-propagation neural network to retrieve the atmospheric temperature and RH profiles after the quality control of the first-level data. However, the traditional back-propagation neural network is time-consuming and requires a huge amount of data [26].

With the continuous development of artificial intelligence technology, the machine learning model has been increasingly applied in the field of microwave remote sensing, especially in atmospheric profile inversion. Gregori et al. [27] used the Gradient Boosting Machine (GBM) regression tree in a machine learning algorithm to estimate the boundary layer height using the MWR data and confirmed the excellent performance of machine learning in terms of training speed and retrieval accuracy. Jia [28] used the Extreme Gradient Boosting (XGBoost) machine learning model to predict the non-monsoonal winter precipitation over Eurasia. The results show that the XGBoost model performs significantly better than the traditional linear regression model. Liu [29] used the XGBoost model to correct the daily land surface temperature, where the rapidly increasing trend after the correction indicates an effective correction of the inhomogeneous land surface temperature in China. Recent studies have shown that the XGBoost model has great potential to improve climate prediction. The Random Forest (RF) algorithm has been applied to atmospheric environmental research in recent years [30,31]. Jiang et al. [32] used the RF machine learning model to establish an aerosol optical depth (AOD) dataset in the cloudy Sichuan Basin. GBM, XGBoost and RF all use boosting learning. The disadvantage of boosting learning is that there is a serial relationship between its base learners, and it is difficult to train data in parallel. The Deep Learning (DL) model is a machine learning algorithm that uses multi-layer artificial neural networks to achieve state-of-the-art accuracy in many tasks [33,34]. Similar to traditional machine learning algorithms, the DL model can model complex nonlinear systems. Moreover, it performs better in extracting the advantageous features with deeper network layers [35]. Recently, the performance of the DL model has been proven to be comparable to that of human experts [34].

In this study, four machine learning algorithms, namely the GBM, XGBoost algorithm, RF algorithm and DL algorithm, are used to compare with the MWR-derived first-level BT data. Based on this, the best machine learning method to improve the retrieval accuracy of profiles from the MWR data will be found. We try to give the evolution of RH profile transfer information, such as in which layer the water content could surge to a certain level, as an indicator of the coming precipitation. The techniques are only tested over a small region in Guangzhou, China. The rest of this paper is organized as follows. Section 2

describes the four machine learning algorithms, the datasets, the study region and the data preprocessing procedures. The comparison between the results of the four machine learning algorithms and the radiosonde data are presented in Section 3, where typical cases are also analyzed. Finally, the conclusion and discussion are presented in Section 4.

## 2. Data and Methods

The microwave radiometer BT data and the sonde data are observed from 2018 to 2021 in Huangpu and Qingyuan, respectively. Machine learning-based models are applied to retrieve temperature and relative humidity using the brightness temperature measured by the microwave radiometer based on the channel from 22.24 GHz to 31.4 GHz, 51.0 GHz to 58.0 GHz. The 2018–2020 dataset is used as the training sample, while the 2021 dataset is used as the validation sample.

### 2.1. Location of Observation Stations

The location of the observation stations is shown in Figure 1. The MWR data are observed from Huangpu station, while the radiosonde data are observed from the Qingyuan station. The distance between Huangpu station (113.29°N, 23.13°E) and Qingyuan station (113.05°N, 23.43°E) is about 70 km. The altitudes of Qingyuan station and Huangpu station are 79.2 m and 70.7 m, respectively. The two stations have similar underlying surface conditions, and there is no mountain barrier between them. The radiosonde data observed at Qingyuan station are well matched with the MWR BT data, which can be applied to train the machine learning algorithms in this study.

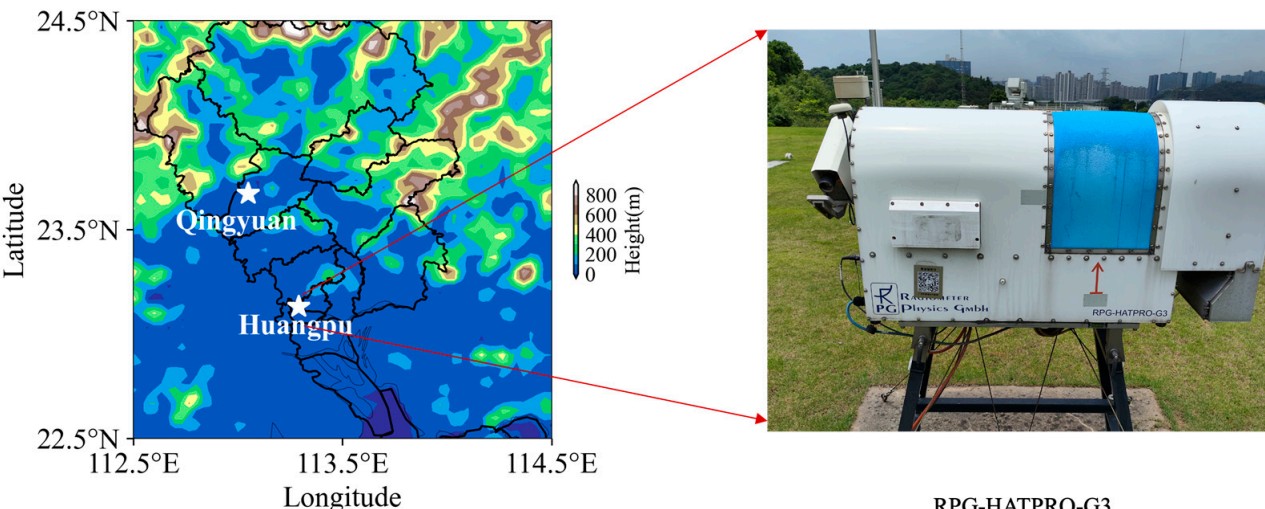

RPG-HATPRO-G3

**Figure 1.** The location of observation stations. (The microwave radiometer data are observed from Huangpu station, while the radiosonde data are observed from the Qingyuan station).

### 2.2. Datasets

Two datasets, namely the MWR BT dataset and radiosonde dataset are used in this paper. The BT data are measured by an MWR located at the Huangpu station. The MWR uses the RPG-HATRPO-G3 from the Radiometer Physics GmbH in Germany, which is a 14-channel ground-based passive MWR with seven water vapor absorption channels (K-band) from 22.24 GHz to 31.40 GHz and seven oxygen absorption channels (V-band) from 51.00 GHz to 58.00 GHz [21,36]. The radiosonde data of temperature and RH are measured by an L-land GTS1 digital radiosonde at the Qingyuan station, which is launched twice daily at 11:00 and 23:00 UTC, respectively.

*2.3. Data Preprocessing*

The quality control of the BT data is performed before the training of four machine learning methods, in order to obtain better prediction results. The datasets during 2018–2021 are matched according to the principle of time consistency. There is a certain law of the time series variation in brightness temperature [20], and the transformation observed by each frequency channel of the MWR within 3 min should be continuous. First, the MWR BT samples are excluded if they do not meet the above conditions. Second, the MWR BT samples are further excluded if they are with large fluctuations, which may be errors of the instrument. After these two sample screening steps, the radiosonde data, temperature and RH are averaged by altitude. The interval altitude for a given level is set to be in the range of every 25 hPa and 50 hPa for below and above 700 hPa, respectively. In this way, the BT data are matched to the radiosonde level values. The matched samples are then classified into three categories based on RH values: clear sky, cloudy sky and rainy conditions. The inversion effect for MWR data is generally better under clear-sky conditions than under cloudy and rainy weather conditions [37–39]. The radiosonde data are processed according to the approach of Yan et al. [21] to determine the weather conditions and estimate cloud parameters. Theoretically, when a cloud forms, the RH at the corresponding height reaches 100%. However, due to factors such as condensation nuclei, the RH in the cloud layer is slightly lower than the theoretical value [40,41]. Therefore, 85% RH is used as the threshold value in the radiosonde data to determine the altitude level. The specific determination criteria are as follows:

(1) The measured data are classified as rainy-day data if the RH is greater than 85% from the ground to the height of 600 m.
(2) The data are classified as cloudy-sky data if the RH is less than 85% near the surface but greater than 85% in the upper atmosphere [20].
(3) The data are classified as clear-sky data if the RH is always less than 85% from the ground to any altitude level.

Thereafter, a total of 2461 quality-assured MWR data samples matched to the radiosonde sounding data from January 2018 to July 2020 are used for training, and 1321 quality-assured test samples during 2021 are used for validation. Due to the cloudy condition, 52% of the three years of data cannot be used.

*2.4. Methods*

2.4.1. Deep Learning (DL)

DL is an advanced machine learning architecture based on neural networks [42]. It aims to bridge the gap between machine learning and artificial intelligence by incorporating powerful learning capabilities and a wide range of applications. Compared to RF, GBM and XGBoost, DL shows superior adaptability. However, DL's performance is highly dependent on the available data, and it tends to excel when provided with a larger volume of data [42]. Recent research indicates that DL exhibits promising results in temperature retrieval [43,44]. The DL model in the study consists of two hidden layers, each containing 200 neurons. The activation function employed in the first hidden layer is relu, as it has been observed to provide superior results compared to other functions [45]. The second hidden layer and the output layer default to using linear activation functions by default. The entire model is trained using the Adam optimizer with the mean square error serving as the loss function.

2.4.2. Gradient Boosting Machine (GBM)

GBM is a boosting algorithm that uses different weights to linearly combine the base learners to reuse the learner with excellent performance [46]. The GBM algorithm calculates the pseudo-residuals according to the initial model. Then, it builds a base learner to interpret the pseudo-residuals, which can reduce the residuals in the gradient direction. Then, the base learner is multiplied by the weight coefficient and linearly combined with the original model to form a new model. The learning rate of the base learner is set to

0.1. The goal of the GBM is to find a model that minimizes the expectation of the loss function [47].

### 2.4.3. Extreme Gradient Boosting (XGBoost)

XGBoost is a gradient boosting-based integrated learning algorithm proposed by Chen and Guestrin [48]. The XGBoost [48] has been gradually applied in the atmospheric environment prediction. A second-order Taylor expansion is introduced in XGBoost, which increases the accuracy and enables loss functions to be customized via gradient descent. It adds the complexity of the tree model to the regularization term in order to prevent overfitting and, as a result, performs better in generalization [49]. However, the applications of the XGBoost machine learning method for the retrieval of meteorological profiles using MWRs are still few [21]. In this study, each tree is constructed using a learning rate of 0.3, a maximum tree depth of 6, a regularization weight of 10 and a total of 50 weak learners (trees).

### 2.4.4. Random Forest (RF)

RF is an integrated machine learning method that uses the random resampling technique bootstrap and the random node splitting technique to build multiple decision trees by Breiman [50]. A splitting technique with 50 random nodes is used to build the model. RF can be used for classification, clustering and regression data applications [51]. The RF model can analyze the classification features of complex interactions and has good robustness to data with noise and missing values. Meanwhile, it has a fast learning speed. The variable importance measure can be used as a feature selection tool for high-dimensional data [52]. Random forests are generally more effective at solving classification problems than regression problems. This is because random forests produce discrete outputs for classification tasks, rather than continuous outputs for regression tasks. In regression, the Random Forest model is limited in its ability to predict values beyond the range of the training set data. Therefore, when performing regression with a Random Forest, it is important to be aware that predictions may be limited within the range of the training data.

### 2.4.5. A 10-Fold Cross-Validation Method

Cross-validation is a common approach to model building and model parameter verification in machine learning, which is used to estimate the skill of a machine learning model [53]. In $k$-fold cross-validation [54,55], it is first randomly divided into $k$ mutually exclusive subsets of similar size, i.e., $k-1$ is randomly selected as the training set each time, and the remaining 1 is used as the test set. When this round is completed, $k$ copies are again randomly selected to train the data. After several rounds (less than $k$), the loss function is selected to evaluate the optimal model and parameters. In this study, $k$ is set to 10. The four machine learning models are trained using 70% of the training samples, following standard training procedures, while the remaining 30% of the samples are used for validation. In addition, 10-fold cross-validation [32] is used for all four models.

## 3. Results and Case Illustration

The study first performs a 10-fold cross-verification analysis of training samples to verify the feasibility of the models. Second, the most appropriate methods of temperature and RH are found by analyzing the total scatter density, errors and RMSEs of different height layers of four machine learning methods with sonde data verification. Third, two cases are used to illustrate the results.

### 3.1. A 10-Fold Cross-Validation with Training Samples

In order to evaluate the performances of the four methods on the training dataset for temperature and RH, we use the 10-fold cross-validation method for verification, and the results are shown in Table 1. For temperature, the root-mean-square errors (RMSEs) of the DL, GBM, XGBoost and RF models are 2.32, 2.33, 2.49 and 3.07, respectively. The

temperatures of the 10-fold cross-validation correlation coefficient (CV-$R^2$) of four methods is above 0.97. For RH, the RMSEs of the RF, XGBoost, GBM and DL models are 13.70, 13.72, 14.96 and 17.96, respectively. RH by the RF model shows the highest accuracy, with the sample-based 10-fold cross-validation CV-$R^2$ being 0.72. The results show the performance evaluation of the four machine methods, where the temperature performs better than RH. Compared with other methods, DL is suitable for MWR temperature and RF is better for RH retrieval. In general, the performance of the training result for RH is not so good.

**Table 1.** The 10-fold cross-validation results for four methods in terms of temperature and relative humidity (RH).

|  | Method | RMSE | CV-$R^2$ | MAE |
|---|---|---|---|---|
| Temperature (°C) | DL | 2.32 | 0.98 | 1.73 |
|  | GBM | 2.33 | 0.98 | 1.80 |
|  | XGBoost | 2.49 | 0.98 | 1.81 |
|  | RF | 3.07 | 0.97 | 2.17 |
| RH (%) | DL | 17.96 | 0.53 | 14.04 |
|  | GBM | 14.96 | 0.67 | 11.09 |
|  | XGBoost | 13.72 | 0.72 | 9.49 |
|  | RF | 13.70 | 0.72 | 9.92 |

RMSE: root-mean-square error; CV-$R^2$: cross-validation correlation coefficient; and MAE: mean average error.

### 3.2. Validation of Four Models with the Radiosonde Data

3.2.1. Scatter Density Variation

Radiosonde data are also used for comparison with the four model retrievals. The temperatures of the four models as a function of the radiosonde measurements from all 22 atmospheric vertical layers from 250 hPa to 1000 hPa are shown in Figure 2. The regression equations and coefficients of determination ($R^2$) are given, as well as the number of data points (N = 1321) and the RMSEs. Figure 2a shows that the linear regression relationship between the DL temperature and radiosonde temperature has a slope of 1.0, a y-intercept of 0.17 and minimal fluctuation around the regression line, with an $R^2$ of 0.98 and the lowest RMSE of 2.36 among the four models. As shown in Figure 2b, the linear regression relationship between the GBM temperature and the radiosonde temperature exhibits a slope of 0.99. The $R^2$ is 0.98, and the RMSE is 2.53. Figure 2c shows that the XGBoost model has a slope of 1.0, an $R^2$ of 0.97 and an RMSE of 3.07. The RF model has a slope of 0.95, an $R^2$ of 0.97 and an RMSE of 3.04, as shown in Figure 2d. The results show that the DL model has a high retrieval capability for temperature with an RMSE of 2.36 °C and an $R^2$ of 0.98.

Similarly, the RH of the four models as a function of the radiosonde measurements from all 22 atmospheric vertical layers is shown in Figure 3. The RMSEs of the DL, GBM, XGBoost and RF models are 20.08, 19.45, 19.72 and 19.07, respectively. The lack of independent cloud-related information may contribute to the deviations. The conditions with an RH less than 85% are considered clear-sky conditions in the study. However, in real atmospheric conditions, clouds may form due to the presence of cloud condensation nuclei when RH reaches around 85% [40]. In general, the RMSEs of RH are relatively greater compared with that of temperature.

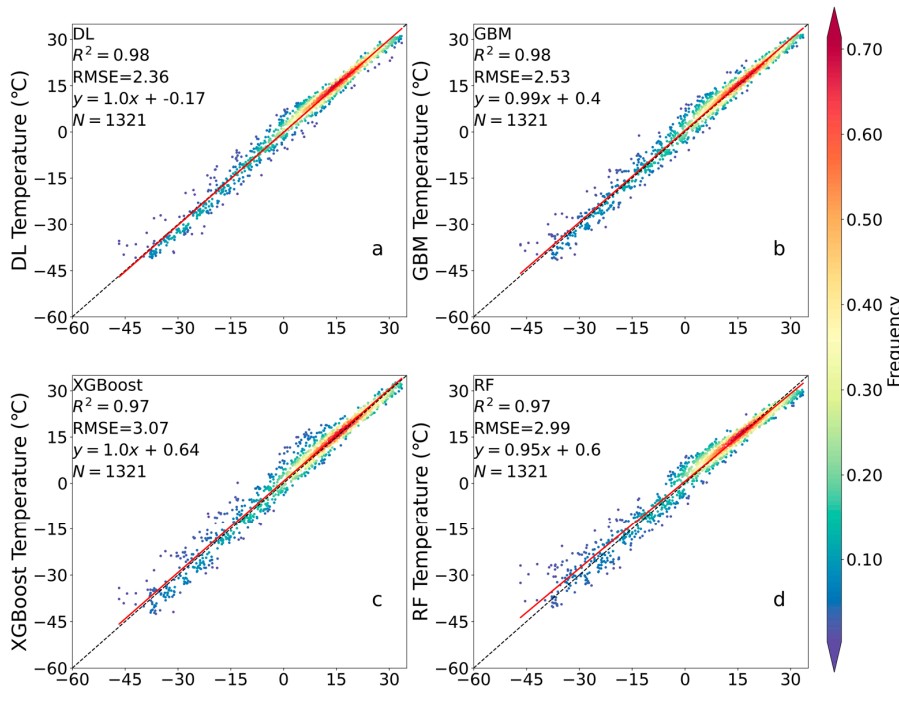

**Figure 2.** Retrievals for temperature as a function of radiosonde data by the (**a**) DL, (**b**) GBM, (**c**) XGBoost and (**d**) RF models. The red solid line is the line of best fit in linear regression.

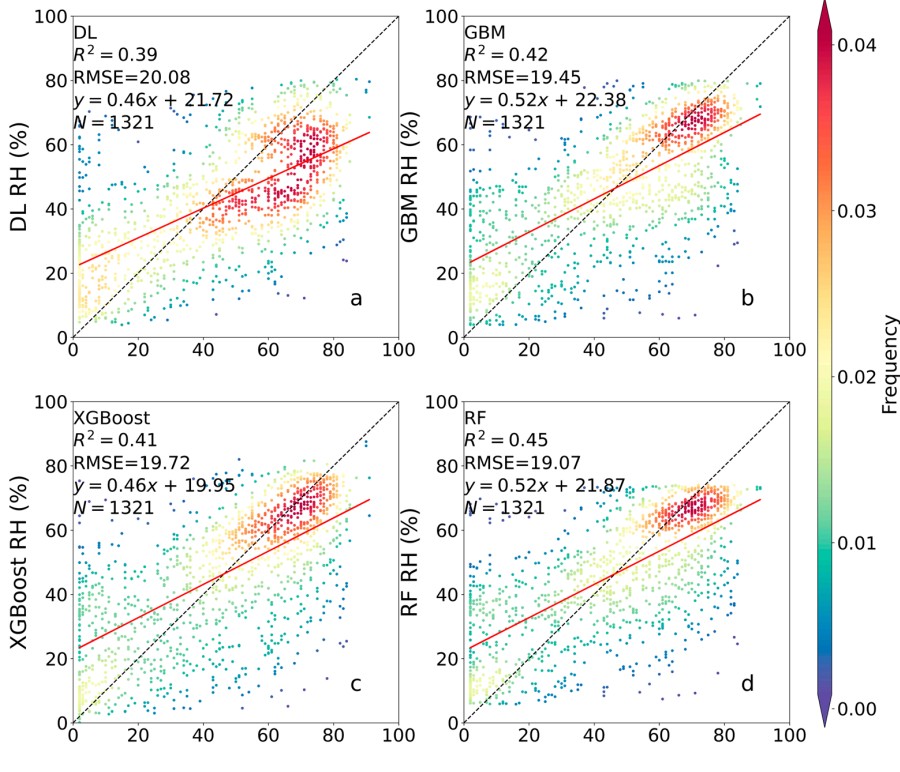

**Figure 3.** Retrievals for RH as a function of radiosonde data by the (**a**) DL, (**b**) GBM, (**c**) XGBoost and (**d**) RF models. The red solid line is the line of best fit in linear regression.

3.2.2. Bias and RMSEs Variation with Altitude

The temperature profile retrieval biases of the four models are shown in Figure 4. The red dotted lines represent the mean value of the biases, and the blue lines inside the box represent the median values. The blue shading indicates ±1 °C temperature biases. The left and right borders of the box contain the values from the first quartile to the third quartile. The blue dotted lines extend from each quartile to the minimum or maximum bias. In Figure 4a, the DL temperature bias is within ±1 °C for most of the pressure levels from 700 hPa to 1000 hPa, indicating high accuracy in retrieving temperature profiles in lower levels. From 250 hPa to 650 hPa, the box length for the DL model is shorter than that of the GBM, XGBoost and RF models, indicating that the temperature biases of the DL model are more concentrated around the mean and median values. Furthermore, most of the mean and median values of the temperature biases by the DL model are very close, indicating that the temperature biases of the DL model are more uniform and concentrated. The temperature bias from 250 hPa to 1000 hPa shows that the mean temperature bias measured by the DL model is negative near the surface and then becomes positive at 850 hPa with the increasing altitude. However, it turns negative again at 350 hPa with the increasing altitude. That is, the temperature bias from the DL model shows a distribution of "low at both ends and high in the middle". In contrast, the temperature biases by the RF model show a large fluctuation from left to right and are not stable enough at all levels (Figure 4d), which is similar to the trend of Yan et al. [21].

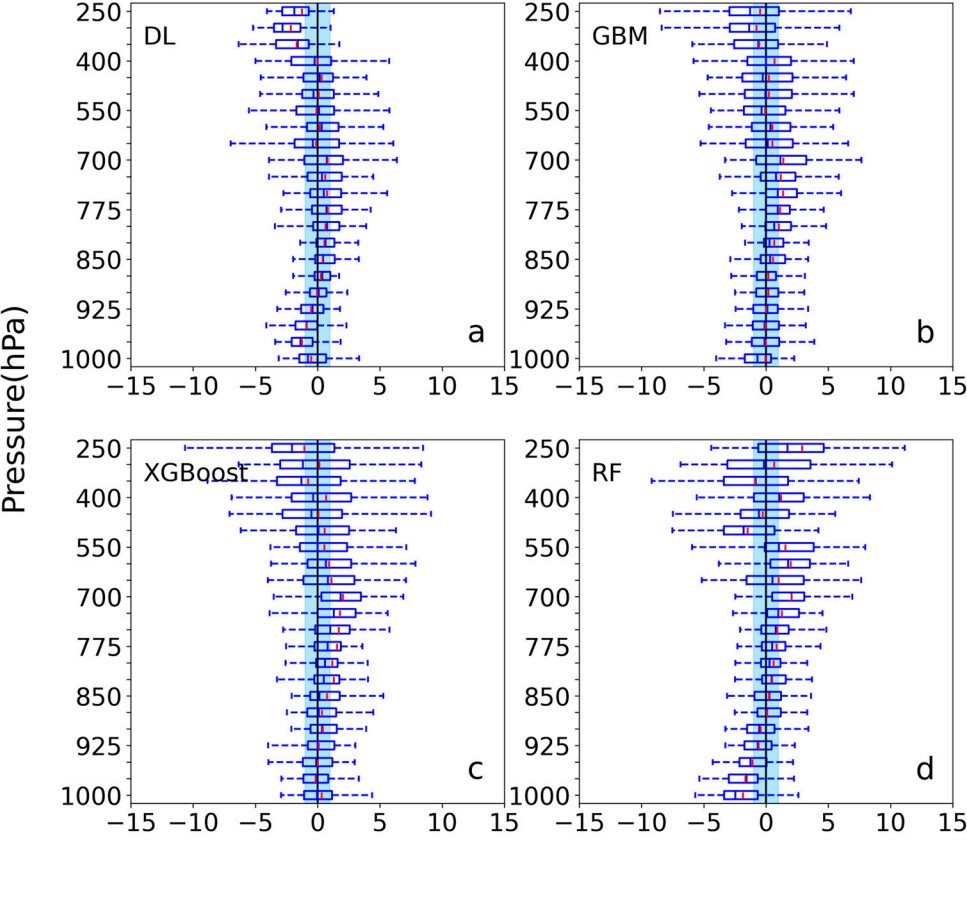

**Figure 4.** Temperature retrieval biases (the retrieval minus radiosonde) by the (**a**) DL, (**b**) GBM, (**c**) XGBoost and (**d**) RF models, respectively. Red dotted lines and blue lines within the boxes indicate the means and medians, respectively. The blue shadow means ±1 °C temperature biases.

　　　　The RH profile retrieval biases of the four models are shown in Figure 5. The red dotted lines and the blue lines are the same as in Figure 4, but the blue shadows show the biases ranging from −10% to +10%. In the four methods, the mean bias is usually close to the median in the lower-altitude layer, but deviates significantly from the median from 600 hPa to 800 hPa. The reason for the large deviation is the loss of the cloud information [18]. Figure 5d shows that the bias of the RF machine learning method remains within ±10% from the surface to 800 hPa, and its box length is almost the smallest among the four models from 800 hPa to 1000 hPa. As for GBM and XGBoost (Figure 5b,c), their bias also remains in the range of ±10% near the surface, but the interquartile range is much larger than the interquartile range of RF and the maximum bias exceeds 40%. Thus, the RF RH shows better retrieval near the surface than the other three methods.

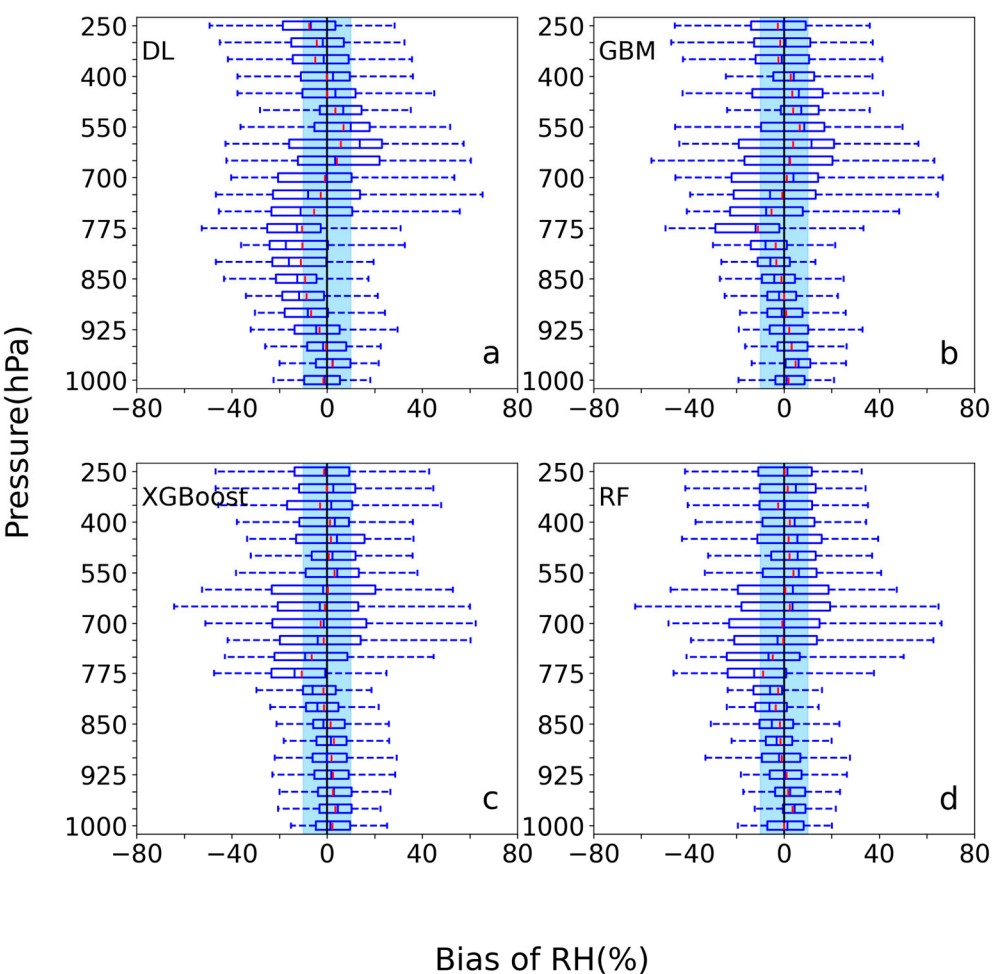

**Figure 5.** RH retrieval biases (the retrieval minus radiosonde) by the (**a**) DL, (**b**) GBM, (**c**) XGBoost and (**d**) RF models, respectively. Red dotted lines and blue lines within the boxes indicate the means and medians, respectively. The blue shadow means a bias of ±10%.

　　　　At the same time, we also compare and analyze the RMSEs of the four methods in different altitude layers in order to find the differences in the performance of the four methods on temperature and RH in different altitude layers. The RMSEs of the temperature and RH profiles of DL, GMB, XGBoost and RF at 22 height layers are shown in Figure 6. For temperature, the DL model shows a smaller RMSE than XGBoost, GBM and RF in the layers from 250 hPa to 1000 hPa. In particular, the DL RMSE is less than 1.5 °C in the layers from 775 hPa to 1000 hPa. For the RH, the RMSE of RH is larger than the RMSE of the temperature, and the RMSEs of the four machine learning methods from 600 hPa to 750 hPa are 20% to 30%; moreover, the RMSEs for all four methods generally increase

with the altitude within this range, and the maximum deviation appears above the level of 700 hPa. This variation characteristic is similar to [20,21]. However, as we can see in the low-level RMSEs performance, the RMSEs of RF are between 10% and 15%. Therefore, from the overall characteristics of the RMSE of the two variables, the temperature RMSE is mainly concentrated in the upper layer, and the high RMSE of RH mainly occurs in the middle layer, which is in accordance with the situation found by Cimini et al. [8]. The results show that the DL temperature from 250 hPa to 1000 hPa performs better than the temperature of the other three methods, and the RF RH performs better in the low layers. In general, the performance of the training result for four machine learning methods is not so good.

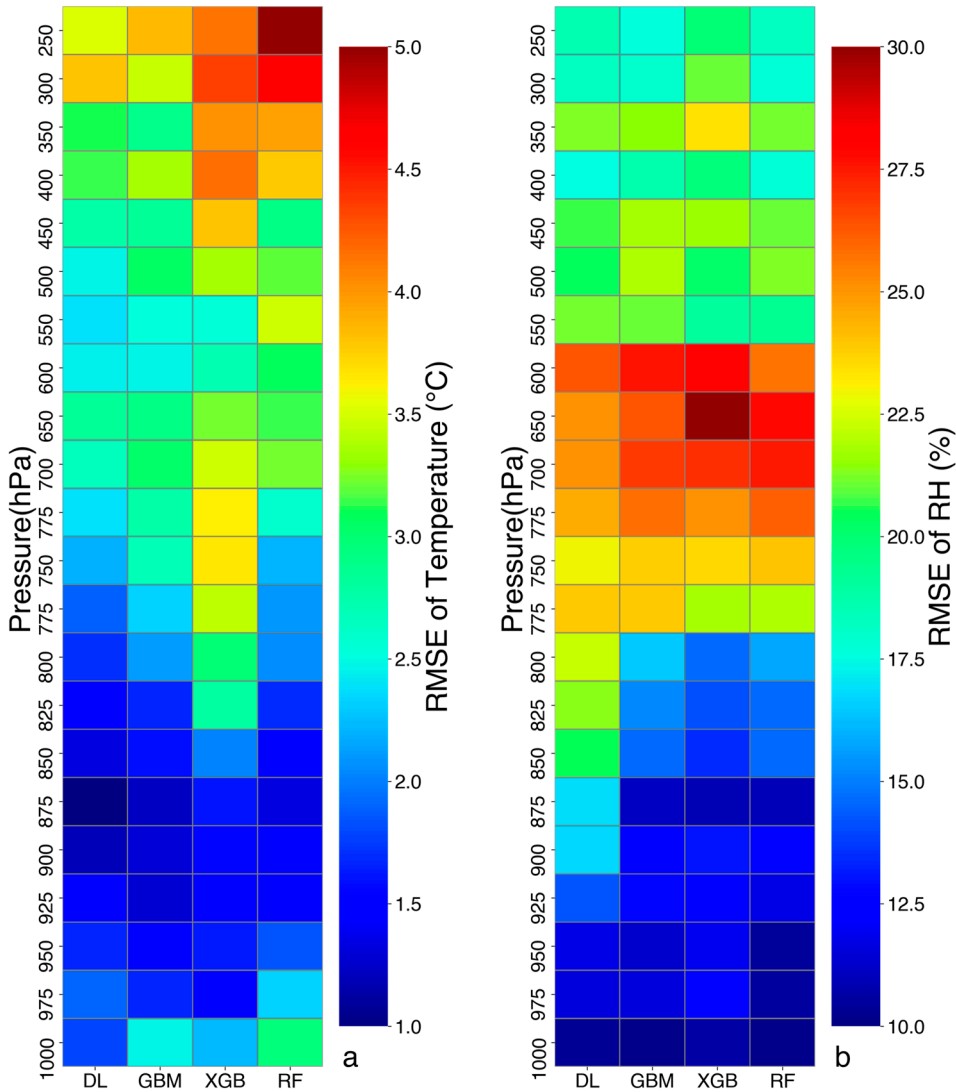

**Figure 6.** RMSEs for retrieval of temperature (**a**), and RH (**b**) profiles relative to the radiosonde data using the DL, RF, GBM and XGBoost methods.

The performance of the four models for RH retrieval at different levels is shown in Table 2. From 900 hPa to 1000 hPa, DL and RF demonstrate better accuracies in terms of $R^2$ (0.62 and 0.60), and the RF model performs better in terms of the RMSE (11.14) and the mean average error (8.83) compared with the DL model (12.93 and 8.84), indicating that the RF RH performs best from 900 hPa to 1000 hPa. From 775 hPa to 875 hPa, the XGBoost model shows better accuracies in terms of an RMSE of 14.97. From 600 hPa to 750 hPa, the RMSEs of the four methods are large. Above 550 hPa, RF performs well for RH retrieval,

with an RMSE of 19.50. According to Table 2, the RF model achieves better retrieval ability from 900 hPa to 1000 hPa and above 550 hPa. From 775 hPa to 875 hPa, the XGBoost model achieves better retrieval ability. The DL model performs well for RH retrieval from 600 hPa to 750 hPa.

**Table 2.** Comparisons among different methods in terms of their performance at different levels.

| Height | Method | RMSE | $R^2$ | MAE |
|---|---|---|---|---|
| 250–550 hPa | DL | 19.69 | 0.34 | 15.09 |
| | GBM | 20.09 | 0.34 | 15.91 |
| | XGBoost | 20.71 | 0.29 | 16.20 |
| | RF | 19.50 | 0.36 | 15.44 |
| 700–750 hPa | DL | 24.79 | 0.27 | 20.65 |
| | GBM | 26.07 | 0.18 | 21.48 |
| | XGBoost | 26.80 | 0.21 | 21.62 |
| | RF | 26.22 | 0.18 | 21.48 |
| 775–875 hPa | DL | 20.99 | 0.18 | 17.73 |
| | GBM | 16.27 | 0.44 | 12.50 |
| | XGBoost | 14.97 | 0.53 | 11.38 |
| | RF | 15.55 | 0.47 | 11.99 |
| 900–1000 hPa | DL | 12.93 | 0.62 | 8.84 |
| | GBM | 11.67 | 0.59 | 9.00 |
| | XGBoost | 12.19 | 0.56 | 9.43 |
| | RF | 11.14 | 0.60 | 8.83 |

Since the RHs of the four machine learning methods show different excellence at different height levels, a new integrated machine learning (ML) RH method is proposed here. The machine learning RH is to select the RH profiles by integrating the four methods of DL, GBM, XGBoost and RF, where the result of the method with the minimum RMSE for a certain level is adopted. The RMSE is 15.00 and $R^2$ is 0.64 by comparison of radiosonde RH and ML RH from all 22 atmospheric vertical levels.

### 3.3. Case Illustration

Based on the analysis in Section 3.2, DL (RMSE = 2.36, $R^2$ = 0.98) is the most suitable for temperature retrieval and ML (RMSE = 15.00, $R^2$ = 0.64) is the most suitable for RH retrieval. A case on 29 January 2021 is used for illustration. Another case with precipitation on 10 February 2021 is used to explain the changes in RH before the entire precipitation.

### 3.3.1. Case Analysis for DL Temperature and Machine Learning RH

Figure 7a shows the temperature profiles from the DL and radiosonde data. At 7:15 a.m. on 29 January 2021 (Beijing time, same below), the DL retrieval shows only a small difference with the radiosonde data from 775 hPa to 1000 hPa. In particular, the DL model agrees well with the radiosonde data at levels from 850 hPa to 900 hPa. However, the difference increases from 250 hPa to 700 hPa. Figure 7b shows the RH profiles by ML. Overall, the difference between the retrieved RH profile and the radiosonde data is greater than that of the temperature profile. The RH obtained from ML and radiosonde data show some agreement with the changing altitude. The integrated ML method performs better for RH from the layers 700 hPa to 875 hPa, with an RH bias lower than 10% compared with other altitudes.

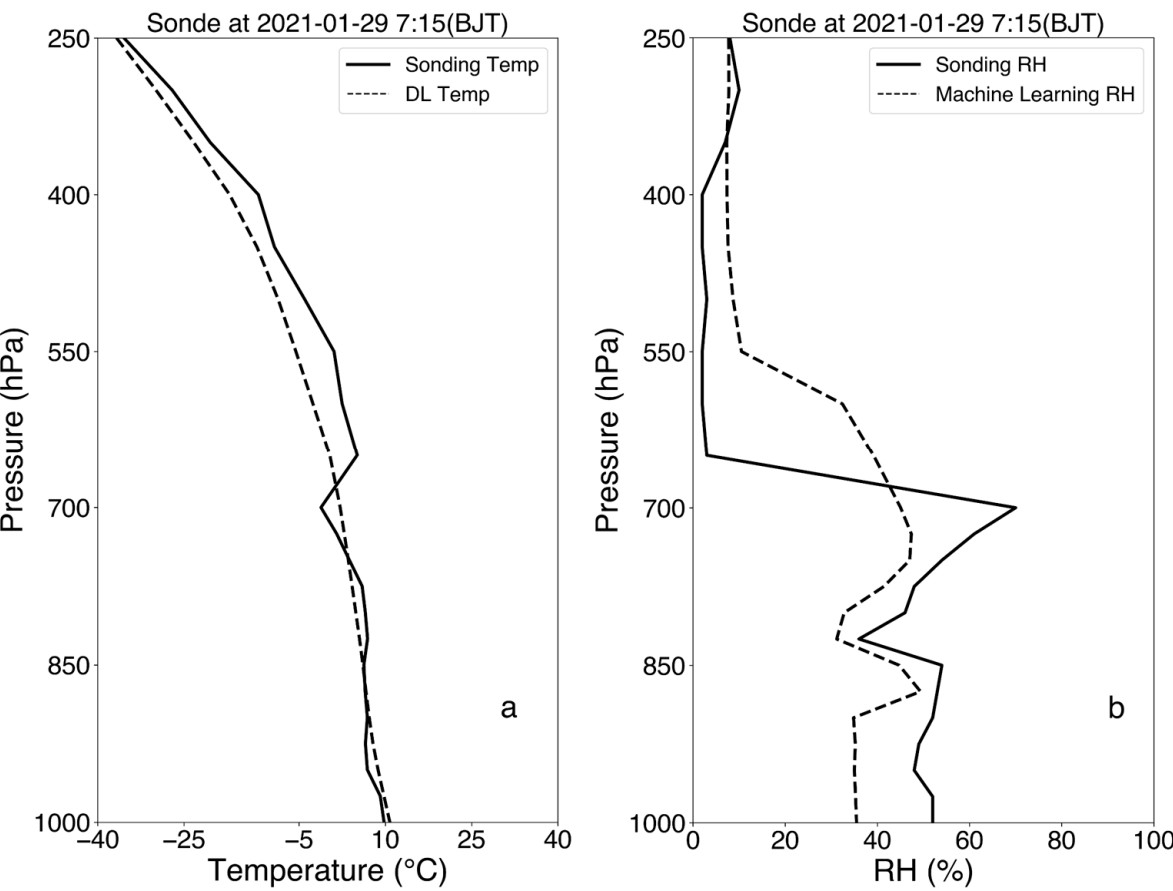

**Figure 7.** Comparison of (**a**) temperature profiles generated using the DL model and radiosonde data, and (**b**) RH profiles generated by integrating machine learning and the radiosonde data at 07:15 on 29 January 2021.

3.3.2. ML RH for Application before Precipitation

Another case on 10 February 2021 is used to explain the changes in ML RH before the entire precipitation process. The vertical profiles of RH predicted from 600 hPa to 1000 hPa levels and the observed data being the liquid water path (LWP) and the hourly surface precipitation histogram before and during a precipitation event are all shown in Figure 8. The precipitation started at 16:00 on 9 February 2021 and ended at 00:00 on 11 February 2021 (Figure 8a). The surface hourly precipitation reached the maximum value (6.9 mm) at 04:00 on 10 February 2021. Two phases of RH changes that occurred before 16:00 on 9 February 2021 are shown in Figure 8. In the first stage, Figure 8a shows the initial peak in the liquid water path (LWP) with a maximum value of 1319.60 g/m², while Figure 8b shows a gradual increase in RH at the lower levels. In the second stage, three consecutive peaks in the LWP (352.60 g/m², 1157.20 g/m² and 885.90 g/m²) were observed, accompanied by an overall increase in RH at all levels. Notably, RH exceeded 85% from 750 hPa to 900 hPa prior to the onset of precipitation. The LWP has four peaks, indicating the continuous moisture transport and humidification process prior to precipitation. It also consists of the RH increase in Figure 8b. The RH increase obtained by the integrated ML method shows good agreement with the pre-precipitation LWP variation curve, indicating that the machine learning-based RH profiles successfully captured the significant increase in humidity before precipitation, which may provide some indication for precipitation forecasting.

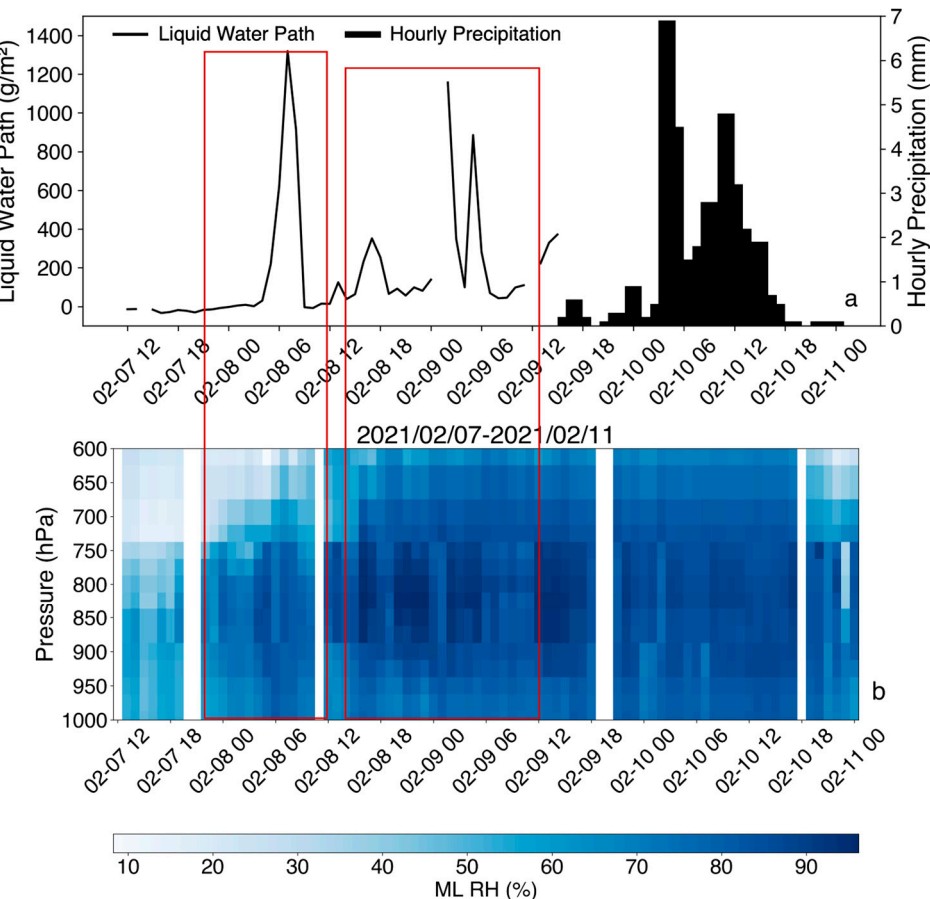

**Figure 8.** (**a**) Observed data of the liquid water path by MWR and the surface hourly precipitation histogram by Huangpu National Basic Meteorological Observation Station from 7 February 2021 to 11 February 2021, and (**b**) vertical profiles of ML RH from 600 hPa to 1000 hPa.

## 4. Discussion and Conclusions

The microwave radiometer (MWR) is widely used in meteorological observations, and the accuracy of temperature and relative humidity (RH) measurements can be affected by retrieval methods, weather conditions and environmental factors. In this study, we compared the Deep Learning (DL), Gradient Boosting Machine (GBM), Extreme Gradient Boosting (XGBoost) and Random Forest (RF) methods in retrieving temperature and RH profiles from 1000 hPa to 250 hPa using the MWR data and radiosonde data from 2018 to 2021, with the aim of improving the profile retrieval accuracy of the MWR.

Validation with radiosonde measurements shows that the DL model has better retrieval capability for temperature with a root-mean-square error (RMSE) of 2.07 °C and a correlation coefficient ($R^2$) of 0.98. Most of the temperature biases in the DL and XGBoost model are within ±1 °C from 700 hPa to 1000 hPa, and the RMSEs of the temperature profile using the DL model are less than 2.5 °C from 750 hPa to 1000 hPa. The RF model performs the best in retrieving the RH with the least bias near the surface, an RMSE around 12.5% and an interquartile range nearly the smallest among the four models.

A new integrated machine learning (ML) RH method is used to select the RH profiles by integrating the four models of DL, GBM, XGBoost and RF, where the result of the model with the minimum RMSE for a certain level is adopted. The RMSE is 15.00 and $R^2$ is 0.64 by comparison of radiosonde RH and ML RH from all 22 atmospheric vertical levels. We use the DL temperature and the ML RH to analyze two cases. A case on 29 January 2021 shows that DL is suitable for temperature retrieval and ML is suitable for RH retrieval. We apply the ML data to a precipitation case on 10 February 2021, and the results show that the change in ML RH shows a close correlation with the liquid water path before 16:00 on 9 February

2021. The ML RH reaches over 85% before 16:00 on February 9, indicating that the machine learning-based RH profiles successfully captured the significant increase in humidity prior to precipitation, which may provide some guidance for precipitation forecasting.

In conclusion, our study provides new insights into the performance of DL, GBM, XGBoost and RF in temperature and RH retrieval using MWR data. DL (RMSE = 2.36, $R^2$ = 0.98) shows superiority in temperature retrieval because the deep neural network architecture allows it to capture complex temperature patterns effectively [33,34]. Similar to traditional machine learning algorithms, DL can model complex nonlinear systems [35]. For the $R^2$ of the temperature, DL and GBM are both 0.98, which is 0.01 higher than that of the RF and XGBoost models. When comparing DL with GBM, the RMSE of temperature decreases from 2.53 to 2.36. The performance of the four models for RH retrieval at different levels is shown in Table 2. The RF model achieves better retrieval ability from 900 hPa to 1000 hPa and above 550 hPa. From 775 hPa to 875 hPa, the XGBoost model achieves better performance. The DL model performs well for RH retrieval from 600 hPa to 750 hPa. An integrated ML (RMSE = 15.00, $R^2$ = 0.64) approach improves RH retrieval because the ML method integrates the advantages of multiple methods.

It is important to note that our training datasets were obtained under clear-sky conditions without considering the data from cloudy conditions, which has certain limitations. Due to cloudy conditions, 52% of the three years of data cannot be used. A total of 2461 quality-assured MWR data samples matched to the radiosonde sounding data from January 2018 to July 2020 are used for training, and 1321 quality-assured test samples during 2021 are used for validation. Although 52% of the data cannot be used, the amount of available data based on observations is enough to represent the Guangzhou area. When applying these machine learning models in another region, it is critical to consider the region-specific characteristics and climatic conditions. The performance of the models may vary due to the differences in atmospheric dynamics, topographies and local weather patterns. Therefore, further investigation and validation specific to the target region would be necessary to assess the suitability and performance of the models. Our stations are only representative of the Guangzhou region. However, the models can be applied in other regions upon the observation available. Yan et al. [21] used microwave radiometer bright temperature data combined with DL, RF and XGBoost to invert the temperature and humidity profile under clear-sky conditions in Beijing. Bao et al. [24] used MWR BT data with a neural network to invert the temperature and humidity profile under clear-sky conditions in Nanjing. The lack of cloud information led to larger errors in RH in the middle layer from 700 hPa to 750 hPa. As shown in Li et al. [17] and Bao et al. [24], the correlation between the RH derived from the MWR and radiosonde data is much smaller than the correlation of temperature. This also proves our conclusion that the temperature inversion results are better than the RH inversion results. In addition, the distribution of the RMSE for all four models from 1000 hPa to 250 hPa is similar to the results of Che et al. [20], which showed that the RH RMSE tends to increase with height and the maximum deviation occurs in the middle atmosphere. Therefore, in this study, the ML method after the fusion of four models is proposed to reduce the influence of nonlinear relations on RH inversion. When the predicted hourly RH reaches the threshold of 85%, the warning information is provided to the forecasters. Consideration of region-specific characteristics is essential, and future research should explore cloud-related analysis. The suitability of DL for temperature retrieval and the effectiveness of ML for RH retrieval can be attributed to their respective model architectures and approaches. Future research will explore more in-depth cloud-associated analyses to address these limitations.

**Author Contributions:** Investigation, Y.L.; formal analysis, Y.L.; writing, Y.L.; visualization, Y.L.; editing, Y.L., H.W. and M.J.; conceptualization, H.W. and M.J.; supervision, H.W. and M.J.; data curation, T.G., H.Y., G.W. and L.Z.; writing—review and editing, Z.W., D.P. and P.T. All authors have read and agreed to the published version of the manuscript.

**Funding:** This study was funded by the Guangzhou Science and Technology Bureau (No. 202206010016), National Natural Science Foundation of China (Grant No. 42105073), Chengdu University of Information Technology Research Fund (KYTZ202217) and the China Scholarship Council.

**Data Availability Statement:** Not applicable.

**Conflicts of Interest:** The authors declare no conflict of interest.

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
