# Peer review of "Machine Learning Model-Based Retrieval of Temperature and Relative Humidity Profiles Measured by Microwave Radiometer"

_remotesensing, doi:10.3390/rs15153838_

Round 1
Reviewer 1 Report
Review of a manuscript titled “Machine Learning Model-Based Retrieval of Temperature and Relative Humidity Profiles Measured by Microwave Radiometer” Yuan Luo et al.
The paper discussed Machine learning-based model retrieval of temperature and relative humidity profiles measured by microwave radiometer. Did the authors apply ML-based models for the retrieval of temperature and relative humidity using microwave radiometer observations? It needs to be clarified from the manuscript that the authors used brightness temperature measured by radiometer for training the model. Or did they use temperature and relative humidity profiles for training models? If later is the case, it should be noted that the microwave radiometer is already trained by Artificial neural networks (ANN) using nearby radiosonde measurements. There is no point in applying the ML model to the ANN outputs again. I suggest authors clearly mention it in the manuscript and recommend it for major revision.
Further, authors have used nearby radiosonde measurements (~ 70 km away from Huangpu) for validation of ML-based models. Again, these radiosonde measurements might have already been used in the microwave radiometer ANN algorithm to produce the temperature and relative humidity profiles.
In addition, the radiosonde may drift, which depends on the season and wind speed. Given the direction, it may approach or away from the microwave radiometer station. If it is away, how can authors justify the results? I suggest including a spatial map of the seasonal wind vector.
Figure 2: In all the models, the linear fit is used, and the slope is found to be nearly 1. Does it mean that the linear fit derived from the seasonal mean is enough than the ML base models? To confirm this, authors should perform the following simple steps: Estimate the seasonal mean (2018-2019) and then compare it with 2020. Fit polynomial and based polynomial coefficients, estimate the temperature and relative humidity and compare with the observations. Based on these simple steps, quantify how much improvement can be obtained with ML-based models.
Figure 3: why the ML-based model simulated RH show large deviations? What could be the reason for this?
Page 2: L64-65. …. Essential for a better forecast. Include proper citation
Page 3: Authors should briefly discuss the radiosonde preprocessing methodology in the manuscript.
Page 4:L140-141: Did the authors verify this statement? Show the figure and take the timings of RH value (>85%) and independent rain rate information.
Page 5: Section 2.4.5: How did the authors choose the k-fold to be 10? How did the authors choose 70% for training? Is it based on random selection? What is the impact of the length of training (60 % or 80%) data on the results? Show the learning curve for the ML models used in the manuscript.
Page 7: L 269-27: How cloud information contributes significant deviation in model RH?
Page 9: 287: How have the authors compared the model-estimated RH and temperature with radiosonde observations? Did they take layer averaged from radiosonde? Include all the details in the methodology section.
Figure 7: Show two more case examples.
Font size is different in the 2.4.1 and 2.4.5 sections. Follow the font size as per the journal format.
Artificial neural networks (ANN) are increasingly used for the retrieval of geophysical parameters from measured brightness temperatures (for example Del Frate et al, 1999; Solheim et al, 1999; Churnside et al; 1994).
We use a standard feed forward neural network [Jung et al., 1998] where the cost function is minimized employing the Davidon-Fletcher-Powell algorithm. The architecture of the ANN used for the retrieval includes an input layer consisting of simulated brightness temperatures for the PRG-HATPRO frequencies, a hidden layer with a certain number of neurons (nodes) and an output layer with the atmospheric variable of interest (LWP, IWV, temperature, or humidity profile). To derive the weights between the nodes of the different layers we generated a data set comprising about 15,000 possible realizations of the atmospheric state, which was divided into three sub sets; the first for training, the second for generalization (finding the optimum number of iterations to avoid over fitting), and the third for evaluating the retrieval RMS. For each output parameter the optimal network configuration – number of nodes in the hidden layer, number of iterations and initial weight – was derived and the retrieval performance was evaluated using the third data subset.
The data set is based on atmospheric profiles of temperature, pressure and humidity measured by radiosondes.
In order to analyze profiles of cloud liquid water content (LWC) from the radio soundings, we chose a relative humidity threshold of 95 % as a threshold for the presence of clouds and calculated a modified adiabatic LWC-profile as proposed by Karstens et al. [1994].
It should be noted that a limitation to ANN algorithm, as to all statistical algorithms, is that they can only be applied to the range of atmospheric conditions, which is included in this data set.
When extrapolations beyond the states included in the algorithm development are made, ANNs can behave in an uncontrolled way, while simple linear regressions will still give a reasonable, although erroneous, result.
Quadratic regressions offer the robustness of a linear regression retrieval with the advantage to model nonlinearities much better than linear regressions. In many cases where unusual atmospheric conditions are likely the quadratic regression is the best choice.
Are the data from the MWRs independent of the radiosondes?
No, not entirely. The retrievals are based on National Weather Service (NWS) radiosonde data from 1994–1999.
Are the tuning functions still used? No. Use of the so-called tuning functions was discontinued on 6 April 1996. The tuning functions were removed from the SGP CF MWR data that were collected between 950101 and 960409. Data collected before this date have already had these removed by Jim Liljegren, while data after this time window never had the tuning functions applied. By removing the tuning functions, the PWV and LWP retrieved from the microwave radiometer are independent of the radiosondes.
What were the tuning functions? The tuning functions linearly relate model-calculated microwave brightness temperatures (using radiosonde data) to brightness temperatures measured with a microwave radiometer. These were needed to account for imperfections in the microwave absorption model used to develop the retrievals that relate precipitable water vapor and liquid water path to the microwave brightness temperatures. The tuning functions should be independent of the instrument and of the location — they should depend only on the microwave absorption model used in the calculations.
How were the tuning functions determined? After each sonde launch, the model that computes the integrated vapor from the sonde as well as the microwave brightness temperatures is run automatically by the data system. Jim Liljegren collected all of these modeled and measured brightness temperatures between October 1992 and December 1993, selected those for which the sky was clear (that is, for which the RMS variation in the liquid-sensing channel brightness temperature was less than 0.4 K), and calculated a regression for each channel.
Atmospheric profiles of temperature, humidity, wind direction and speed are typically measured by radiosondes launched from facilities maintained by the national weather services
Reviewer 2 Report
The authors evaluated the performance of various machine learning algorithms to retrieve temperature and humidity profiles from radiometer observations. In general, I think the topic of the manuscript has the potential to add novel scientific insights and fits the scope of the journal. However, the current version has, in my opinion, a limited scientific impact. Please find my major concerns below.
1. The explanation of the implemented machine learning models is too brief and too general (even face recognition is mentioned). Please elaborate more on the models in the context of your research and elaborate on the differences between the models.
2. Elaborate on your data: what data is given to the models? i.e. the timestep that is closest to the radiosonde, all brightness temperature observations with equal weights or do the authors distinguish between the different bands? Furthermore, the authors mentioned that the data cleaned and filtered, but don’t mention how they do this. This information needs to be provided.
- The discussion is too short, doesn’t elaborate on the results and does not put the results into context. Missing information and other suggestions regarding the discussion are (numbered 1-4):
- 1. Many manuscripts already discuss ML/DL methods and their performance. This manuscript would provide novel insights if, for instance, it would address why does a certain type of model performs better.
- 2. Furthermore, please elaborate on the effects of your results if you would perform it in another region. Are the models only representative for this type of surface? Furthermore, the authors mention the stations are located at locations with similar characteristics, but how representative are these characteristic for the region?
- 3. Cloudy conditions are not considered. What percentage of the three years of data cannot be used due to this condition? Please add this information to L. 147-149.
- 4. Why is DL more suitable for temperature retrieval and ML for RH retrieval? Please provide thoughts about this in your discussion.
3. The techniques are only tested over a small region in China. I understand that it is not possible to study global retrieval, but it would be good to place your results in context.
4. The conclusion states that the retrieval can be used in precipitation warning systems. However, the model is not tested during cloudy observations. Can the authors please elaborate on this claim and how the model can still give precipitation warnings?
5. Figure 8 is very difficult to interpret, both the upper and lower figure. Please also describe which panel represent the radiosondes and which the radiometer retrieved profiles. How should the reader compare the upper and lower panel? In other words: please elaborate on the message you want to convey with this figure.
Words are sometimes incorrectly used, some sentences are unclear, and the interpunction is not always correct. Advised to adjust the manuscript and to make sure you use the right synonyms. Examples (please note I did not mention all of them, so the authors have to check the entire manuscript): L. 162-164, L. 187-189, L. 201-202 (does ‘methods’ mean models?), L. 354 (consisted must be consistent, I think).
Reviewer 3 Report
1. line59-61, 'Therefore, it is essential to control the quality of the MWR observation data before its application', Please briefly introduce the quality control methods used.
2. line 384, 'Our training data sets are under clear sky conditions, and data under cloudy conditions are not considered in this study'. Since cloud and rainfall data are not involved in machine training, how can machine learning accurately predict precipitation?
Reviewer 4 Report
Review of the manuscript "Machine Learning Model-Based Retrieval of Temperature and Relative Humidity Profiles Measured by Microwave Radiometer" by Yuyan Luo, Hao Wu, Taofeng Gu, Zhenglin Wang, Haiyan Yue, Guangsheng Wu, Langfeng Zhu, Dongyang Pu, Pei Tang, Mengjiao Jiang [Remote Sens. 2023, 15, x FOR PEER REVIEW].
This manuscript presents to us the four machine learning methods and also proposes one additional integrated model retrieval of temperature and relative humidity profiles measured by microwave radiometer and their agreement with the corresponding sounding data. Hereafter, the four-year measurements from Huangpu meteorological station in Guangzhou, China and the radiosonde data from the Qingyuan meteorological station (70 km northwest of Huangpu station) during the 2018-2021 period are used. The results presented here indicate that the DL (Deep Learning) method performs the best in temperature retrieval, while the relative humidity of the four machine learning methods show different excellence at different altitude levels. As an illustration of the quality of the different machine learning methods the authors selected two cases.
This manuscript shows us the importance of introducing the methods of artificial intelligence in meteorological measurements. The quality of the considered methods seems to be very high, especially for the temperature profiles. In my opinion, this study therefore can be published in this form.
Author Response
Thank you for your comment.
We made some revisions to improve the manuscript.
Round 2
Reviewer 1 Report
The authors have addressed all the comments/suggestions. I, therefore, recommend it for publication.
Author Response
Thank you for your comment.
Reviewer 2 Report
I would like to thank the authors for their response and improving their manuscript. The authors did implement some of the provided suggestions. However, some suggestions still received no attention or were wrongly interpreted by the author's. Hence, I recommend the following changes.
1. Include in the manuscript why a certain model performs better
2. The discussion is too brief. Again, I would highly stress to put your results into context by using already published studies.
3. You mention in your response that your results indicate the model could be used elsewhere. But do you have any proof for that? e.g. previous research, another case study.
4. Add the context of the response you gave to my second point regarding the discussion into your manuscript
5. Add something about the representativity of the data if 52% of the data could not be used due to cloudy conditions. Furthermore, if this model only works for non-cloudy conditions, how would it this model work over areas where no information about clouds can be provided?
6. I had the following comment: "The conclusion states that the retrieval can be used in precipitation warning systems. However, the model is not tested during cloudy observations. Can the authors please elaborate on this claim and how the model can still give precipitation warnings?". In your response, your mention that it cannot be used as precipitation forecast or quantification. This only further underlines my point. Please remove this statement from your discussion and explain for which applications this method is relevant.
As a last note: thanks a lot for clarifying figure 8!
Please remove typo's and make sure English words are used in the right context. Only the ones I explicitly mentioned were removed, but as I mentioned in the previous revision, more are found in the manuscript.
Round 3
Reviewer 2 Report
Thank you for your kind responses and adjustments. I think the manuscript is ready to be published.